



# Study of the fractality in an MHD Shell model forced by solar wind fluctuations

Macarena Domínguez[1], Giuseppina Nigro[2], Víctor Muñoz[3], Vincenzo Carbone[2], and Mario Riquelme[1]

[1]Departamento de Física, Facultad de Ciencias Físicas y Matemáticas, Universidad de Chile, 8370449 Santiago, Chile
[2]Dipartimento di Fisica, Universita della Calabria, 87036 Rende CS, Italy.
[3]Departamento de Física, Facultad de Ciencias, Universidad de Chile, 7800003 Santiago, Chile.

**Correspondence:** Macarena Domínguez (mdominguezv@ug.uchile.cl)

**Abstract.** The description of the relationship between interplanetary plasma and geomagnetic activity requires complex models. Drastically reducing the ambition of describing this detailed complex interaction, and if we are interested only in the fractality properties of the time-series of its characteristic parameters, a magnetohydrodynamic (MHD) shell model forced using solar wind data, might provide a possible novel approach. In this paper we study the relation between the activity of

the magnetic energy dissipation rate obtained in one such model, which may describe geomagnetic activity, and the fractal dimension of the forcing.

In different shell model simulations, the forcing is provided by the solution of a Langevin equation where a white noise is implemented. Since this forcing has shown its unsuitableness in describing the driver due to solar wind action, we propose to consider the fluctuations of the product between the velocity and the magnetic field solar wind data as the noise in Langevin

equation, whose solution provides the forcing in the magnetic field equation.

We compare the fractal dimension of the magnetic energy dissipation rate obtained, of the magnetic forcing term, and of the fluctuations of $v \cdot b_z$, with the activity of the magnetic energy dissipation rate. We examine the dependence of these fractal dimensions on the solar cycle. We show that all measures of activity have a peak near solar maximum. Moreover, both the fractal dimension computed for the fluctuations of $v \cdot b_z$ time series and the fractal dimension of the magnetic forcing have a

minimum near solar maximum. This suggests that the complexity of the noise term in the Langevin equation may have a strong effect in the activity of the magnetic energy dissipation rate.

## 1 Introduction

There are many investigations regarding the relation between interplanetary plasma parameters and the occurrence of geomag-

netic events in the Earth's magnetosphere (Kane, 2005; Gonzalez et al., 1994, 2004; Tsurutani et al., 1988; Burton et al., 1975; Rathore et al., 2015; Snyder et al., 1963). Among these, Rathore et al. (2015); Kane (2005) show a decrease of the antiparallel





geomagnetic field, $B_s$, before the occurrence of the minimum of *Dst*. While Gonzalez et al. (1994, 2004); Burton et al. (1975) introduce an energy balance equation where the *Dst* index and the rectified interplanetary electric field ($\mathrm{d}v \cdot B_z$) are related.

The study of the fractal dimension in various fields has allowed to understand diverse phenomena in the light of universal

concepts, adding a new, interdisciplinary perspective to nonlinear systems. For example, this approach has been used to study seismicity, to describe the distribution of epicenter and hypocenters in a given geographical zone (Pastén et al., 2011), or by considering the relationship between the fractal dimension of the spatial distributions of the aftershocks and the faults (Nanjo and Nagahama, 2004; Sahimi et al., 1993); in music, where the characterization of musical pieces has been done through fractal dimensions (Gündüz and Gündüz, 2005; Hsü and Hsü, 1990; Su and Wu, 2007); and in plasma physics, where the use of fractal

dimensions to understand plasma properties is becoming increasingly common (Chang, 1999; Macek et al., 2005; Szczepaniak and Macek, 2008; Chang and Wu, 2008; Neto et al., 2008; Materassi and Consolini, 2007; Zaginaylov et al., 2002; Carreras et al., 2000; Yankov, 1997; Dimitropoulou et al., 2009; McAteer et al., 2010; Domínguez et al., 2014, 2017, 2018).

Fractal dimensions can be calculated from either time series or spatial patterns. For instance, the fractal dimension of the time series of auroral electrojets (AE) or from spatial data such as solar magnetograms, has shown interesting properties,

being generally a non-integer value and less than the Euclidean dimension (Aschwanden and Aschwanden, 2008a, b; Kozelov, 2003; McAteer et al., 2005). Several studies have analyzed the relationship between the fractal and multifractal dimension with physical properties, which has provided a tool to predict events on the surface on the Sun (solar flares), the solar wind, and the Earth's magnetosphere (Dimitropoulou et al., 2009; Aschwanden and Aschwanden, 2008a; McAteer et al., 2010; Uritsky et al., 2006; Georgoulis, 2012; McAteer et al., 2005; Conlon et al., 2008; Chapman et al., 2008; Kiyani et al.,

2007).

There are many different methods to calculate the fractal and multifractal dimensions. In a previous work, we have studied the temporal evolution of solar and geomagnetic activity, by calculating a scatter box-counting fractal dimension from solar magnetograms and *Dst* data (Domínguez et al., 2014). The fractal dimension of the *Dst* analysis decreases during magnetic storms, an effect that is consistently observed across several time scales, from individual storms to a complete solar cycle. Our

results suggest that this definition of fractal dimension is an interesting proxy for complexity in the Sun-Earth system, not only for static data but also when the evolution of solar and geomagnetic activities are followed.

Moreover, in Domínguez et al. (2017), the authors show that the fractal dimension and the occurrence of the bursts in magnetic energy dissipation rate $\epsilon_b(t)$ computed in an MHD shell model integration have correlations similar to those observed in geomagnetic and solar wind data. In that work, the forcing terms of the MHD shell model are provided by the solution of the

50 Langevin equation, where a white noise is employed. That forcing, previously adopted, shows stationary statistical properties, hence revealing its inadequacy to describe the effect of solar wind evolution which is acting on the magnetospheric activity. In order to mimic the evolution of the magnetospheric forcing due to the solar wind, in Domínguez et al. (2018) the MHD shell model has been forced using magnetic and velocity field data measured in the solar wind. This latter work shows a peak in the activity of $\epsilon_b(t)$ near solar maximum, whereas the fractal dimension of the forcing magnetic field time series has a minimum

near solar maximum.



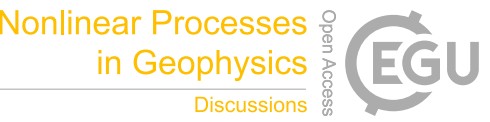
Considering these results, in this paper we present an attempt to describe the complex interaction between solar wind and magnetosphere using a very simple model, where we employ $v \cdot b_z$ data to deduce a suitable forcing for the magnetic field evolution. In particular the fluctuations of $v \cdot b_z$ values inferred from solar wind data are introduced as the noise in the Langevin equation. Then, the solution of this latter equation provides a forcing that we introduce in the magnetic field equation. Thus, by using data that are related to the occurrence of geomagnetic activity, our aim is to investigate whether the statistical properties described in this model evolve because of the evolution in the statistical properties of the forcing term. In particular, in this paper we study whether there is a relationship between the fractality of the forcing on one hand, and the activity and the fractality of the dissipation on the other one.

The paper is organized as follows. In Sec. 2 we present the main features of the MHD shell model used to calculate the magnetic energy dissipation rate $\epsilon_b(t)$, as well as the method used to modify the forcing term of the model. In Sec. 3 we describe the method to calculate the fractal dimension of the fluctuation of $v \cdot b_z$ used as noise term on the Langevin equation. In Sec. 4 we describe the method to calculate the fractal dimension of the magnetic forcing term, and the energy dissipation rate obtained from the shell model. In Sec. 5 we present the definitions of the activity parameters used to analyze the energy dissipation rate. In Sec. 6, the results obtained are presented and finally, in Sec.7 our conclusions are discussed.

## 2   Shell Model

In general, shell models allow to deal with the nonlinear dynamics of fluid systems, reproducing relevant features of MHD turbulence even for high Reynolds numbers, which involve a large computational cost in direct numerical simulations (Boffetta et al., 1999). This is done by means of a set of equations —a simplified version of the Navier-Stokes system— which greatly reduce the available degrees of freedom (Obukhov, 1971; Gledzer, 1973; Yamada and Ohkitani, 1988).

In this work, we use the MHD GOY shell model, which has been shown to be adequate to describe the dynamics of the energy cascade in MHD turbulence (Lepreti et al., 2004), dynamo effect (Nigro and Carbone, 2010; Nigro and Veltri, 2011), statistics of solar flares (Boffetta et al., 1999; Lepreti et al., 2004; Nigro et al., 2004), finite-time singularities in turbulent cascades (Nigro and Carbone, 2015) and to model the fractal features of a magnetized plasma (Domínguez et al., 2017, 2018).

This model is described in more detail in our previous works (Domínguez et al., 2017, 2018). Below we focus on the choice of forcing terms, which is relevant for the present study.

In the model, the wave-vector space ($k$-space) is divided into $N$ discrete shells of radius $k_n = k_0 2^n$ ($n = 0, 1, \ldots, N$). Then, two complex dynamical variables $u_n(t)$ and $b_n(t)$, representing, respectively, velocity and magnetic field increments on an eddy scale $l \sim k_n^{-1}$, are assigned to each shell.

The following set of ordinary differential equations describes the dynamical behavior of the model (Lepreti et al., 2004)

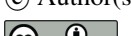


$$\frac{du_n}{dt} = -\nu k_n^2 u_n + ik_n \left\{ (u_{n+1}u_{n+2} - b_{n+1}b_{n+2}) \right.$$
$$-\frac{1}{4}(u_{n-1}u_{n+1} - b_{n-1}b_{n+1})$$
$$\left. -\frac{1}{8}(u_{n-2}u_{n-1} - b_{n-2}b_{n-1}) \right\}^* + f_n \,, \tag{1}$$
$$\frac{db_n}{dt} = -\eta k_n^2 b_n + ik_n \frac{1}{6} \left\{ (u_{n+1}b_{n+2} - b_{n+1}u_{n+2}) \right.$$
$$+ (u_{n-1}b_{n+1} - b_{n-1}u_{n+1})$$
$$\left. + (u_{n-2}b_{n-1} - b_{n-2}u_{n-1}) \right\}^* + g_n \,. \tag{2}$$

Here, $\nu$ and $\eta$ are, respectively, the kinematic viscosity and the resistivity; $f_n$ and $g_n$ are external forcing terms acting, respectively, on the velocity and magnetic fluctuations.

Initially, velocities in the 2nd and 4th shell are set to complex random numbers, whereas the initial magnetic field fluctuations are set to zero, $b_n(t = 0) = 0$. (Domínguez et al., 2017, 2018)

If we set $\nu = \eta = 10^{-4}$ and $N = 19$, consistent with the choices in Domínguez et al. (2017, 2018), and numerically integrate the shell model Eqs. (1)–(2), we can calculate the magnetic energy dissipation rate defined as:

$$\epsilon_b(t) = \eta \sum_{n=1}^{N} k_n^2 \left| b_n^2 \right| \,. \tag{3}$$

In previous work (Domínguez et al., 2017; Lepreti et al., 2004; Nigro et al., 2004; Nigro and Carbone, 2010; Nigro and Veltri, 2011; Nigro, 2013; Nigro and Carbone, 2015) the forcing terms were obtained from the Langevin equation

$$\frac{d\tilde{f}_n}{dt} = -\frac{\tilde{f}_n}{\tau_0} + \mu(t), \tag{4}$$

where $\tilde{f}_n = f_n$ or $g_n$, $\tau_0$ is a correlation time introduced in a Gaussian white noise $\mu$ of width $\sigma$. This provides a stochastic way to drive turbulence in the model. However, turbulence in space plasmas is not always subject to stationary drivers. Such is the case, for instance, of the Earth's magnetosphere. This system is driven by the solar wind, which itself has its own dynamics on short time scales due to local events such as CMEs, and on longer time scales as the solar cycle. In this paper we deal with

105 this property of the drivers, by considering a non-stationary forcing of the shell model.

One possible way of characterizing the stationarity of the forcing given by Eq. (4) is by calculating its fractal dimension, which is a simple measure of the complexity of the time series. Following Domínguez et al. (2014, 2017), a scatter plot is built from the time series, and then the box-counting fractal dimension of this plot is calculated and associated to the time series. When this method is applied to the output of Eq. (4) in various time windows, a value of $D \sim 1.7$ is obtained. Its independence

of the used time window is a manifestation of the stationary character of the time series.

A first method to change the fractality of the forcing terms was presented in Domínguez et al. (2018). In that case, from two scalars —the flow speed and the average magnetic field of the solar wind as obtained by





OMNI (https://cdaweb.gsfc.nasa.gov/istp_public/), two complex series $f_1$ and $g_1$ were built, which were used as
forcing of the first shell in Eqs. (1)–(2). In this way, it was shown that the activity of the resulting $\epsilon_b(t)$ time series has a peak
near solar maximum. However, the magnetic field time series seems to be the most sensitive, because its fractal dimension
seems to correlate with the solar cycle much more than the fractal dimension of the velocity field time series. In fact, the latter
does not show any particular sensitivity to the solar magnetic activity.

We now explore a second possibility to change the fractality of the forcing terms, namely, changing the method to calculate
the stochastic term for magnetic forcing, which corresponds to $\mu(t)$ in Eq. (4). The conventional method to solve this equation,
as mentioned above, considers $\mu(t)$ as a white Gaussian noise (Nigro et al., 2004; Lepreti et al., 2004).

Usually the forcing is applied only on the velocity equation, while in Domínguez et al. (2017) this forcing is considered for
both the velocity and magnetic field equation.

Here, we will preserve the Gaussian noise for the velocity field, while for the magnetic field forcing we will use the fluctua-
tions in $v \cdot b_z$,

$$\mu(t) = v \cdot b_z - \langle v \cdot b_z \rangle, \tag{5}$$

where $v$ and $b_z$ are the velocity and $z$-component magnetic field of the solar wind, respectively. This difference between the
velocity and magnetic field forcing is because, as mentioned before, the velocity time series does not show any relation with
the solar magnetic activity (Domínguez et al., 2018).

The data of the solar wind used in this work are obtained from the OMNIWeb Plus data an service
(https://cdaweb.gsfc.nasa.gov/istp_public/). We consider this source because OMNI is a compilation of data ob-
tained from many space missions (IMP 8, Geotail, Wind and ACE) of the magnetic field of the solar wind near the Earth. More
specifically, we use data of $v$ and $b_z$ at 1 AU of distance with 1 minute of resolution. The coordinate system of the data are
the Geocentric Solar Ecliptic, GSE. Thus, the $z$-axis is the projection of the axis of the Earth's magnetic dipole (positive to the
north) on the plane perpendicular to the $x$-axis (towards the Sun).

Given that the forcing term must be a complex number in Eq. (2), a random phase $\varphi$ is needed for each datum that is
calculated from equation (4) for the magnetic case. Then,

$$f_b(t) = \tilde{f}_b(t)e^{2\pi i\varphi}, \tag{6}$$

where the amplitude $\tilde{f}_b(t)$ corresponds to the solution of the Langevin equation using the modified $\mu(t)$, and $f_b(t)$ is the force
term used in the shell model code.

In order to account for the variability with solar activity, we generate 13 time series of the magnetic energy dissipation, each
one using the data corresponding to the 13 years of the 23th solar cycle (1996 to 2008). Once the time series are generated, we
define four indices to measure the activity of the magnetic energy rate ($\epsilon_b$) and analyze the relationship between these and the
fractality of the data.





## 3 Box-counting dimension of the fluctuation of $v \cdot b_z$

In this work, we use the same definition as in Domínguez et al. (2014, 2017) for the scatter plot box-counting fractal dimension. The fractal dimension for each time series of the fluctuation of solar wind data [see Eq. (5)] is estimated from their scatter diagram. If $\bar{\mu}^i$ is the $i$-th $\bar{\mu}$ datum in the series, and $\bar{N}$ is the total number of data, the scatter diagram is a plot of $\bar{\mu}^{1+(i+1)j}$ versus $\bar{\mu}^{1+i \cdot j}$, for $0 \leq i \leq (\bar{N}-1)/j$ and with $j$ integer.

Then, the scatter diagram is divided in square cells of a certain size $\varepsilon$, and we count the number $\bar{N}(\varepsilon)$ of cells which contain

a point. Next, we consider several values of $\varepsilon$, and we find the range of $\varepsilon$ where $\log(\bar{N}(\varepsilon))$ scales linearly with $\log \varepsilon$. If the slope in this region is given by $-D_j$, then in this region,

$$\bar{N}(\epsilon) \propto \varepsilon^{-D_j}. \tag{7}$$

Figure 1 illustrates the three steps to calculate the fractal dimension, using data for year 2000. Two values of $j$ are used as an example, $j = 1$ and $j = 10$.

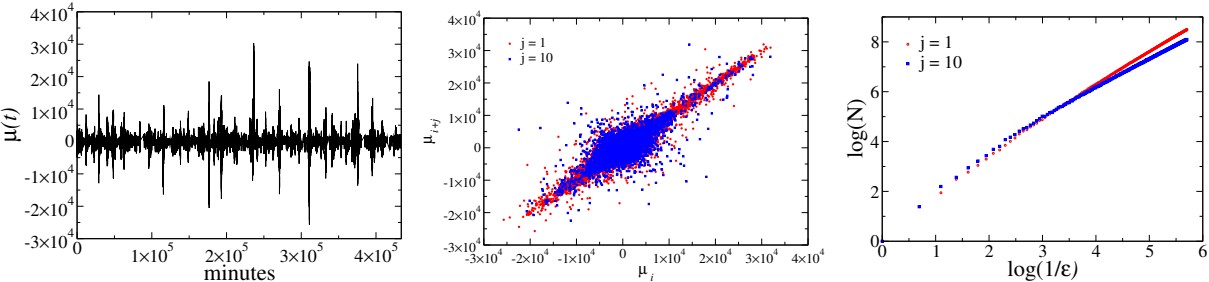

**Figure 1.** Left: $v \cdot b_z - \langle v \cdot b_z \rangle$ for the year 2000. Center: Scatter diagram. Right: log-log plot of Eq. (7). Results for two values of the data sampling are shown: $j = 1$ (red points) and $j = 10$ (blue points)

## 155 4 Box-counting dimension of the magnetic forcing term and the energy dissipation rate

We used the same definition as in the previous section to calculate the fractal dimension of $\tilde{f}_b(t)$ and $\epsilon_b(t)$. Using one year-data of $v \cdot b_z$ fluctuations as magnetic field forcing term, we obtain a $\tilde{f}_b(t)$, and an energy dissipation rate time series. Then, for a given data window in this series, we construct the scatter plot and calculate its box-counting fractal dimension as described in Sec. 3.

Figure 2 illustrates these three steps to calculate the fractal dimension of the $\epsilon_b$ time series. We can see that due to the very high time resolution in our computer simulation, necessary to properly solve the shell model equations, the change in $\epsilon_t(t)$ at each iteration is very small, leading to a scatter plot for $j = 1$ which is essentially a straight line of slope 1, and thus to a box-counting dimension equal to 1 as well. However, for larger values of $j$ the scatter diagram presents a nontrivial structure.


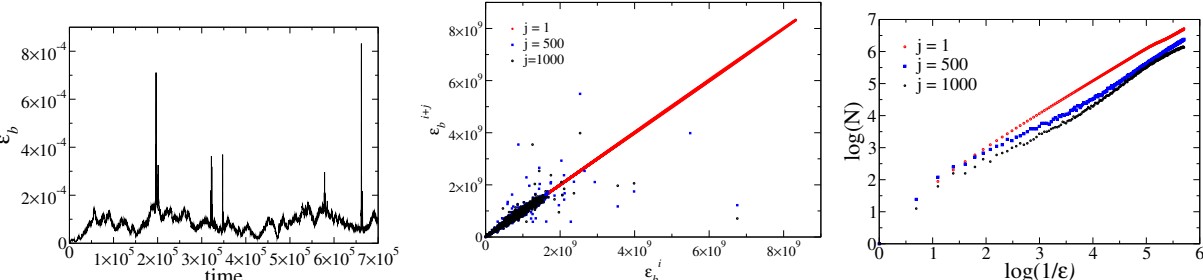

**Figure 2.** Left: $\epsilon_b$ for the year 2000. Center: Scatter diagram. Right: log-log plot of Eq. (7). Results for three values of the data sampling are shown: $j = 1$ (red points), $j = 500$ (blue points), and $j = 1000$ (black points).

## 5  Activity parameters

Some studies have reported a variation in the (multi)fractal features of the solar wind within the solar cycle (Szczepaniak and Macek, 2008; Macek, 2006, 2007; Macek and Wawrzaszek, 2009). On the other hand, it is well established that geomagnetic activity increases during solar maximum, and various models attempt to correlate specific features of the solar wind with geomagnetic activity (Gonzalez et al., 2004; Rathore et al., 2014; Kane, 2005; Huttunen et al., 2002; Rangarajan and Barreto, 2000; Echer et al., 2004). In this section we investigate whether the amount of complexity in the shell model forcing (as
measured by its fractal dimension) somehow correlates with the level and complexity of the dissipation activity.

To this end, we need to define activity parameters for the output time series. We use the same parameters as in Domínguez et al. (2018). First, a threshold $\tilde{\epsilon}$ is chosen, so that an "active state" is said to appear whenever $\epsilon_b(t) > \tilde{\epsilon}$. Then, four activity parameters are defined:

– $N$: is the number of data above that threshold.

– $\langle\epsilon_b\rangle$: is the average of the data.

– $\langle\epsilon_b\rangle_{\mathrm{up}}$: is the average of the data above the threshold.

– $\max(\epsilon_b)$: Is the maximum value of $\epsilon_b$.

The threshold was defined as:

$$\tilde{\epsilon}_b = \langle\epsilon_b\rangle + n \cdot \sigma, \tag{8}$$

where $\sigma$ is the standard deviation of the time series and $n$ is a number between 1 and 10. (In Domínguez et al. (2017) only $n = 5$ and $n = 10$ were considered.)





# 6 Results

We first study the fractal dimension of the noise term used to solve the magnetic Langevin equation, namely $\mu(t)$. In general, the fractal dimension of the time series is expected to depend on the value of the time delay $j$. Thus we study its dependence
on $j$, as well as its dependence on the solar cycle. Results are shown in Fig. 3.

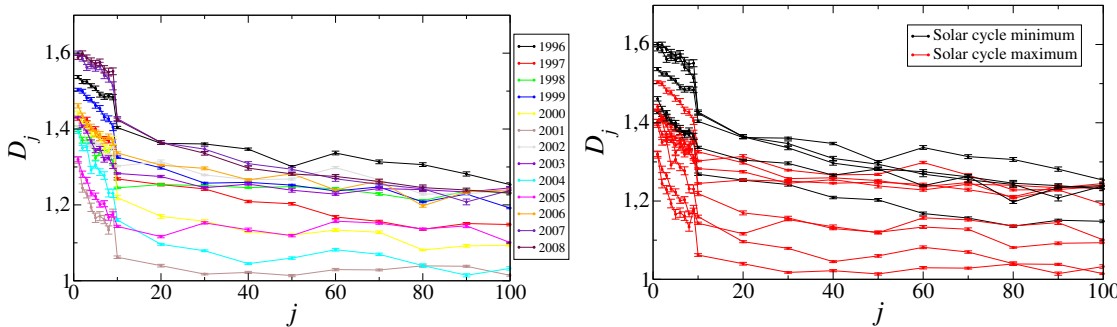

**Figure 3.** Box-counting dimension of $\mu(t)$ for different values of $j$. Left: Curves for each year of the 23rd solar cycle. Right: curves are distinguished for years corresponding to maximum (black lines, years 1998 to 2005) or minimum (red lines, years 1996, 1997, 2006 to 2008) of the solar cycle.

The left panel of Fig. 3 shows that there is a general trend for the fractal dimension to decrease with $j$. Previous results based on the shell model simulation (Domínguez et al., 2017) show that active and quiet states can be distinguished by their different behavior with $j$, and thus Fig. 3 may mean that the fractal properties of the time series does not depend on the stage of the solar cycle. However, in the same Domínguez et al. (2017) it was shown that the behavior may be different for low, intermediate, and
190 large values of $j$. Since $j$ in turn is related to the data sampling rate, another possibility is that the universal behavior in Fig. 3 is a consequence of the time resolution of the data. With the data in this paper, it is not possible to discriminate between these two options.

However, although the trend with $j$ is similar for all years, the values of the fractal dimensions are not. In order to show that, we take sunspot number data obtained from National Geophysical Data Center, prepared by U.S. Dept. of Commerce, NOAA,
Space Weather Prediction Center (SWPC) (ftp://ftp.swpc.noaa.gov/pub/weekly/RecentIndices.txt). By inspection, we notice that the yearly average number of sunspots near solar minimum is below 40, whereas it quickly increases above 50 when approaching solar maximum, reaching 178 in 2002. Thus we set $N_s = 40$ as the threshold. If the number of sunspots in a year is greater that $N_s$, the year is classified as closer to solar maximum; if the number of sunspots is less than $N_s$, it is classified as closer to solar minimum. With this criterion, years 1996, 1997, 2006–2008 are classified in the minimum, and
200 years 1998–2005 in the maximum of the solar cycle. The result is shown in the right panel of Fig. 3, where we clearly see that, on average, the fractal dimension of the fluctuations $\mu(t)$, discriminates between solar cycle minimum and maximum curves.

On the other hand, in the right panel of Fig. 3, we can see that when $j$ increases, the distinction between the minimum and maximum years improves. Then, henceforward we use $j = 100$ to illustrate our findings.


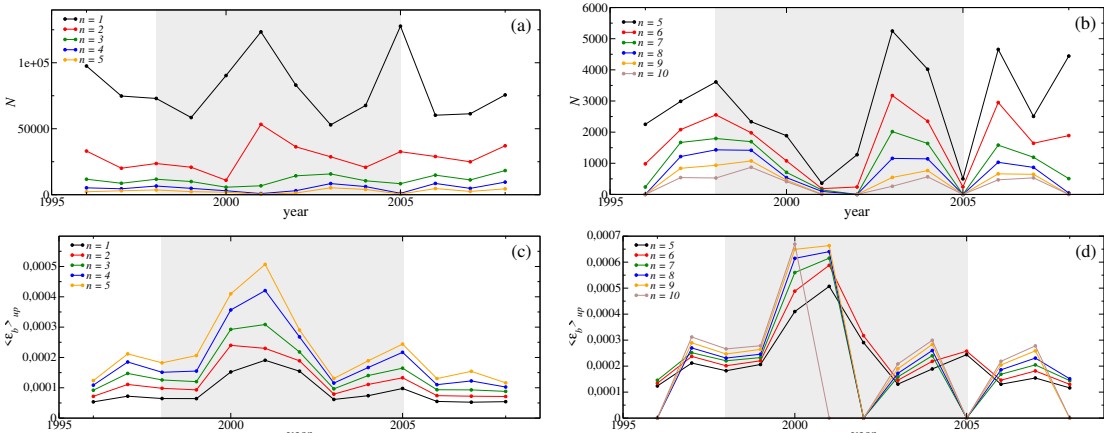

**Figure 4.** $N$ and $\langle \epsilon_b \rangle_{\mathrm{up}}$ calculated from the 13 time series of $\mu(t)$ for different values of $n$. For clarity, separate plots are shown for two sets of values for $n$: from 0 to 5 [panels (a), (c)], and from 5 to 10 [panels (b) , (d)]. Grey region corresponds to years of maximum solar activity, as described in the text.

We intend to compare the degree of complexity of the input time series with the level of activity in the dissipated magnetic
energy. We have proposed four ways to measure activity in Sec. 5. For each year of the 23rd solar cycle, the solar wind fluctuation time series $\mu(t)$ is used to force the shell model, and the resulting activity in the output is measured.

As stated in Sec. 5, there are two parameters of the activity that depend of the threshold $\tilde{\epsilon}_b$. With the aim to select the appropriate value of $n$ in Eq. (8), in Fig. 4 we first show the results for two of the activity parameters $N$ and $\langle \epsilon_b \rangle_{\mathrm{up}}$. We note that for $n = 5$, $\langle \epsilon_b \rangle_{\mathrm{up}}$ has a clear peak near the solar maximum [year 2002, Fig. 4(c)]. If the threshold is too large [Fig. 4(d)],
sometimes no data are found above it, and the activity parameter drops to zero. In the case of $N$ [Figs. 4(a,b)], the curves for all values of $n$ yield similar results. No curve has a clear maximum near solar maximum, suggesting that this parameter is rather insensitive to solar activity. Notice that in Domínguez et al. (2018), $N$ was the parameter that showed the strongest correlation with the solar cycle. This highlights the complexity in the definition of a suitable metric for activity. On the other hand, Fig. 4 shows that the model does respond to various activity levels in the forcing time series, whether such forcing involves the fields
themselves (Domínguez et al., 2018) or their fluctuations (this work).

Based on the previous discussions, we conclude that a moderate value of $n$ is appropriate when defining the activity parameters, so that the anomalous behavior in Figs. 4(b,d) is avoided. We will take $n = 5$.

We now compare the various activity parameters with the fractal dimension of $\mu(t)$. Figures 5 and 6 show the fractal dimension of $\mu(t)$, and one of the activity parameters for each time series. We see that, in general (except for $N$) the maximum
of the activity parameters computed for $\epsilon_b(t)$, approximately occurs in the years around the solar maximum. Moreover, the fractal dimension of $\mu(t)$ decreases during the same period.

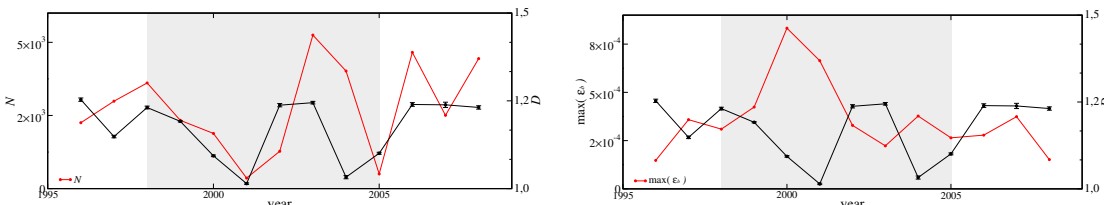

**Figure 5.** Box-counting dimension of $\mu(t)$ for the solar wind with $j = 100$ (black line), and the activity of the dissipated magnetic energy $\epsilon_b(t)$ (red line), as measured by the parameter $N$ (left panel) and $\max(\epsilon_b)$ (right panel). Threshold to define activity is $n = 5$ [see Eq. (8)]. Grey region corresponds to years of maximum solar activity, as described in the text.

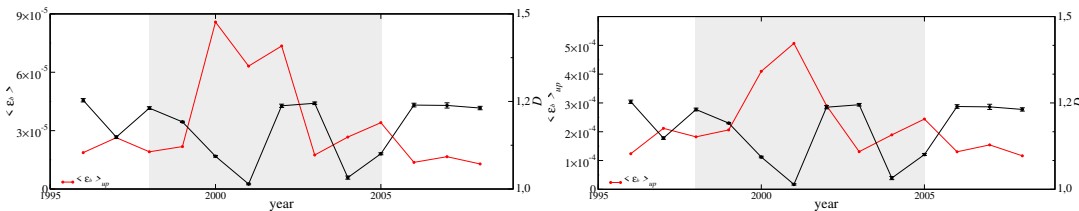

**Figure 6.** Same as Fig. 5, but for activity parameters $\langle \epsilon_b(t) \rangle$ (left panel) and $\langle \epsilon_b \rangle_{up}$ (right panel). Grey region corresponds to years of maximum solar activity, as described in the text.

We perform a similar analysis, but for the fractal dimension of the magnetic forcing term, $\tilde{f}_b(t)$ in Eq. (6). That is, we calculate the dependence of the fractal dimension on $j$ for the $\tilde{f}_b(t)$ time series, and relate it to the stage within the solar cycle as was done in Fig. 3. Results are shown in Fig. 7.

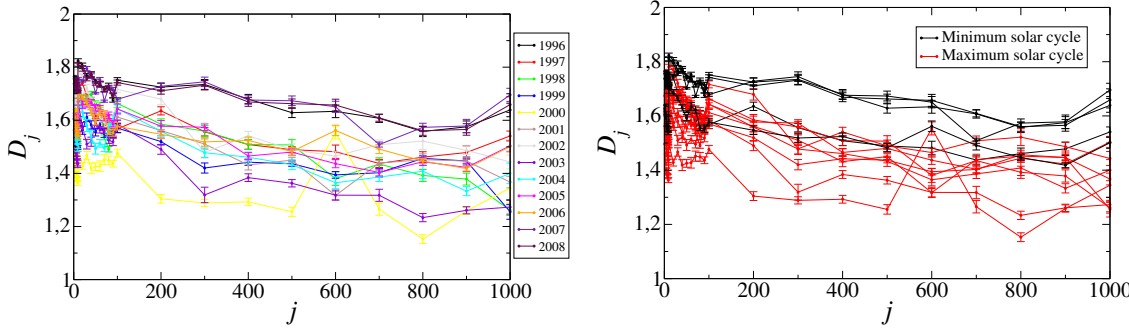

**Figure 7.** Box-counting dimension of magnetic forcing term $\tilde{f}_b(t)$ for different values of $j$. Left: For all years of the 23th solar cycle. Right: curves are distinguished for years corresponding to maximum (black lines, years 1998 to 2005) or minimum (red lines, years 1996, 1997, 2006 to 2008) of the solar cycle.

The conclusion from Fig. 7 is similar to the one deduced in the previous analysis for $\mu(t)$ (Fig. 3). The general trend of the fractal dimension is to decrease with $j$, and the fractal dimension of the magnetic forcing during years close to solar minimum

is, in general, larger than the one measured during the years near solar maximum. In the right panel of Fig. 7, this is more clear for $j > 300$.

Considering the above results, we now choose $j = 500$ for the following figures. In Figs. 8 and 9, we compare the fractal

dimension of the magnetic forcing term $\tilde{f}_b(t)$ with $j = 500$ for each year, with the same activity parameters of Figs. 5 and 6. We can see that, like the previous analysis, the fractal dimension of the magnetic forcing term has a minimum in the years of maximum activity. Also, consistent with Figs. 5 and 6, results for $N$ are less clear, as seen in Fig. 8.

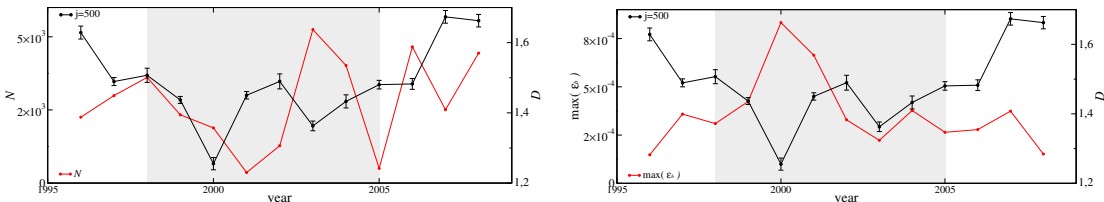

**Figure 8.** Box-counting dimension of magnetic forcing term for $j = 500$ with respective activity of $\epsilon_b(t)$ (red lines): $N$ (left) and $\max(\epsilon_b)$ (right), with $n = 5$. Grey region corresponds to the maximum period of the solar cycle, as described in the text.

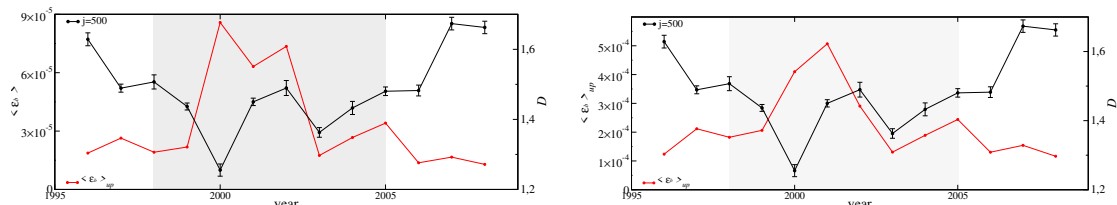

**Figure 9.** Box-counting dimension of magnetic forcing term for $j = 500$ with respective activity of $\epsilon_b(t)$ (red lines): $\langle \epsilon_b(t) \rangle$ (left) and $\langle \epsilon_b \rangle_{up}$ (right), with $n = 5$. Grey region corresponds to the maximum period of the solar cycle, as described in the text.

Finally, we perform the analysis of the fractal dimension of the magnetic energy dissipation rate $\epsilon_b(t)$ for different values of $j$. This latter fractal dimension, depicted in Fig. 10, does not show any particular dependence on the solar cycle, at least during

the 23rd solar cycle here considered.

It is important to note that for small values of $j$, the fractal dimension is essentially constant. In fact, for $j = 1$ the fractal dimension is always one for all years, due to the scatter diagram being exactly a line (see figure 2). Unlike Fig. 7, Fig. 10 does not suggest a robust correlation between the fractal dimension of $\epsilon_b(t)$ and the solar cycle, for any value of $j$.

Therefore, the time-dependent fractal dimension that characterized the forcing here adopted, leads to noticeable variations

in the intermittency of the magnetic energy dissipation rate, as measured by the activity parameters above defined. On the other hand, the same quantity, namely magnetic energy dissipation rate, does not show any significant variations of its fractal dimension during the cycle considered.


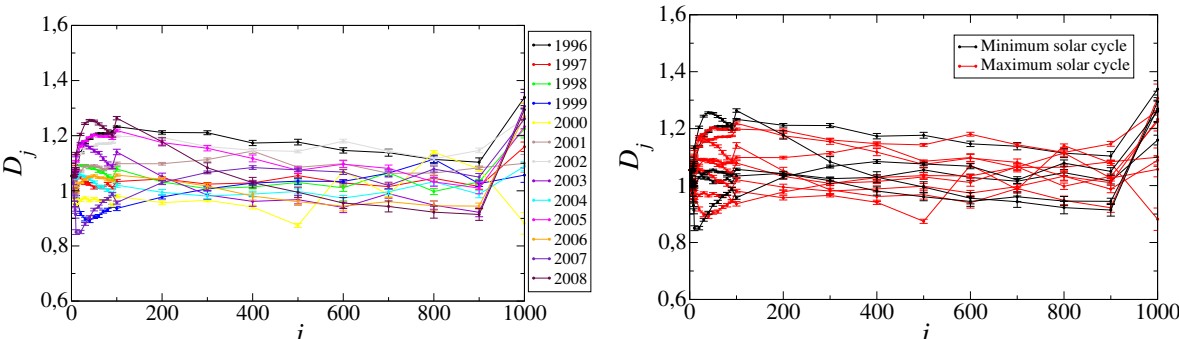

**Figure 10.** Box-counting dimension of energy dissipation rate for different values of $j$. Left: For all years of the 23th solar cycle. Right: curves are distinguished for years corresponding to maximum (black lines, years 1998 to 2005) or minimum (red lines, years 1996, 1997, 2006 to 2008) of the solar cycle.

## 7 Conclusions

In this paper we present the results of an MHD shell model where we force differently the velocity field fluctuations and the magnetic field fluctuations. In particular, while the forcing employed in the velocity equation is a time-correlated Gaussian noise, for the magnetic field equation we adopt the solution of a Langevin equation where the fluctuations of $v \cdot b_z$, computed using solar wind data, are introduced in this equation instead of a stochastic term. This is accomplished in order to have a forcing on the magnetic field equation that mimics the time-dependent solar wind action on Earth's magnetosphere during a solar cycle. This description is certainly an oversimplification of the complex dynamics that determines the interaction between solar wind and Earth's magnetosphere, but it provides a possible approach if we are interested only on the fractal properties of characteristic parameter time series.

In this framework, we have analyzed the relationships of the activity of the magnetic energy dissipation rate obtained in the shell model, with the fractal dimension of its input and output time series. Specifically, our defined activity parameters are compared with: the fractal dimension of the fluctuations of the solar wind $v \cdot b_z$ data, the magnetic force term, and the time series of the magnetic energy dissipation rate.

Both the fluctuation term, $\mu(t)$, and the resulting forcing term [Eqs. (5) and (6)] have a fractal dimension which is well correlated with the solar cycle, as shown in Figs. 3 and 7, indicating that information on solar activity is actually present in the fractal dimension of $\mu(t)$ and resulting forcing. This is not the case for the magnetic energy dissipation rate, as can be seen in Fig. 10. Thus, this complexity measure produces signatures of the corresponding solar activity when applied to the input of the shell model, but does not produce them when applied to the output of the model.

For the quantities which possess a time-dependent fractal dimension, namely $\mu(t)$ and the forcing term, this dimension exhibits a minimum near solar maximum. As to the activity of the output, all proposed metrics —except $N$— seem to correlate with the solar cycle, showing a peak near the solar maximum. This suggests that the complexity of the noise term of Langevin





equation may have, within the simulation, a noticeable effect in the activity of the magnetic energy dissipation rate, although

the fractal dimension, as calculated here, is not a suitable metric for that output activity.

*Competing interests.* The authors declare that they have no conflict of interest.

*Acknowledgements.* We thank the support of CONICYT through FONDECYT Grant No. 1121144 (V.M.). and No. 3160305 (M.D).





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
