# Peer review of "Study of the fractality in an MHD Shell model forced by solar wind fluctuations"

_Nonlinear Processes in Geophysics, 2019_

## Referee Comment (RC1) · Anonymous Referee #1 · 8 Oct 2019

General comments

The authors provide an interesting extension of their research on fractal dimensions generated from their modified MHD GOY shell model. In particular, the authors choose the forcing term in the velocity Langevin equation as a Gaussian noise while for the magnetic Langevin equation the forcing term is generated by the observed solar wind fluctuations of the velocity and the z-component of the magnetic field. The fractal dimensions of the magnetic energy dissipation rate, the magnetic forcing term and the magnetic energy forcing rates are then compared and conclusions are made from these comparisons.

The presentation of the method used, the history of the development of the magnetic shell model, and the estimates of the fractal dimensions are all clearly described. And

their final results are reasonably explained.

Specific comments

[1] There were no specific explanations for the choice of the set of parameters: nu, eta and N.

[2] Magnetic dissipation rate is calculated but not the velocity dissipation rate. Why?

---

## Referee Comment (RC2) · Anonymous Referee #2 · 29 Oct 2019

Dear Editor,

I have reviewed the manuscript (MS) "Study of the fractality in an MHD Shell model forced by solar wind fluctuations" by Dominguez et al. submitted for possible publication in *Nonlinear Processes in Geophysics (NPG)* and found that it may be acceptable for publication in *NPG* following intermediate revisions. This MS presents a comparison between the fractal dimension of the activity of the magnetic energy dissipation rate obtained in a magnetohydrodynamic (MHD) shell model, which may describe geomagnetic activity, and the fractal dimension of the solar wind forcing expressed through the $v \cdot bz$ fluctuations. However, there is clearly a lack of (a) a discussion of the significance of the herein presented results, along with (b) a comparison with previous studies that simultaneously investigated the complexity between geomagnetic activity and solar wind variations, which point to necessary revisions for this study.

Sincerely,

*Comments*

1. Introduction
   Other interesting applications of fractal dimension analysis on systems related to geophysical and biological extreme events can be found in Eftaxias et al. (2006, 2008).

   l. 48: please define the abbreviation "MHD" here.

2. Section 2
   l. 75: please define the abbreviation "GOY" here.

3. Results I
   In the right panel of Figures 3, 7 and 10 the fractal dimension curves are provided for the 13 years of solar cycle 23. However, the discrimination between curves corresponding to years of solar minimum and curves corresponding to years of solar maximum is not straightforward. In Figure 3 a couple of curves of solar minimum are mixed with the curves of solar maximum. Why? What are the years of solar minimum represented by these curves? Figure 7 discriminates between curves of solar minimum and maximum in a clearer way and Figure 10 does not discriminate at all. Why is that?

4. Results II
   Figures 5 and 6 compare the variations of fractal dimension between solar wind data and dissipated magnetic energy: the fractal dimension of magnetic energy is higher than the fractal dimension of *VBsouth* during solar maximum. What's the interpretation of this observation? IMHO, it might mean that the complexity of the magnetosphere is higher than the complexity of the corresponding solar wind variations during solar maxima. Why is that? Back in 2006 there has been a study comparing the variations of the Hurst exponent

(i.e., another complexity measure) between a geomagnetic activity index, indicating magnetic storm intensity, and *VBsouth* variations (Balasis et al., 2006). This was done for 2001 a year around solar cycle maximum. It was found that while the magnetosphere is mostly driven by the solar wind the critical feature of persistency (lower complexity) in the magnetosphere is the result of a combination of solar wind and internal magnetospheric activity rather than solar wind variations alone. In that study the degree of complexity between Dst and *VBsouth* was apparently different. In the present study the same seems to hold for the dissipated magnetic energy and *VBsouth*. Please comment upon the past and present findings.

5. Conclusions

Please provide motivation / rationale for performing this study. What are the differences / improvements between the present study and recent studies on the topic by the same group of authors (e.g. Dominguez et al., 2018)? What is the significance of the obtained results for the Space Weather community? Eventually, please compare your results with other pertinent studies (e.g. Donner et al., 2019).

*References*

Balasis G., I. A. Daglis, P. Kapiris, M. Mandea, D. Vassiliadis, and K. Eftaxias (2006), From pre-storm activity to magnetic storms: a transition described in terms of fractal dynamics, Ann. Geophys., 24, 3557-3567.

Domínguez, M., Nigro, G., Muñoz, V., and Carbone, V.: Study of the Fractality of Magnetized Plasma using an MHD Shell Model Driven by Solar Wind Data, Phys. Plasmas, 25, 092 302, https://doi.org/10.1063/1.5034129, 2018.

Donner, R. V., Balasis, G., Stolbova, V., Georgiou, M., Wiedermann, M., & Kurths, J. (2019). Recurrence-based quantification of dynamical complexity in the Earth's magnetosphere at geospace storm timescales. Journal of Geophysical Research: Space Physics, 124, 90–108. https://doi.org/10.1029/2018JA025318.

Eftaxias K., Kapiris P., Balasis G., Peratzakis A., Karamanos K., Kopanas J., Antonopoulos G. and Nomicos K. (2006), A Unified Approach to Catastrophic Events: From the Normal State to Geological or Biological Shock in Terms of Spectral Fractal and Nonlinear Analysis, Natural Hazards & Earth System Sciences, 6, 205 – 228.

Eftaxias K., Y. Contoyiannis, G. Balasis, K. Karamanos, J. Kopanas, G. Antonopoulos, and C. Nomicos (2008), Evidence of fractional-Brownian-motion type asperity model for earthquake by means of pre-seismic electromagnetic emissions, Natural Hazards & Earth System Sciences, 8, 657 – 669.

Potirakis, S. M., G. Minadakis, and K. Eftaxias, Sudden drop of fractal dimension of electromagnetic emissions recorded prior to significant earthquake, Nat Hazards, 2012, 64, 641–650, doi: 10.1007/s11069-012-0262-x.

---

## Author Comment (AC1) · 26 Dec 2019

Please find enclosed the comments to the Referee concerning the paper titled "Study of an MHD Shell Model forced with Solar Wind data". The Referee's comments are in bold, and our answers in regular font. When- ever applicable, the modified portions of the text are included in italics in our response. We have also addressed the editorial comments regarding the figures. Finally, we would like to thank the Referee's comments. We think the paper is much improved thanks to them.

Regards,

Macarena Dominguez

Please also note the supplement to this comment:
https://www.nonlin-processes-geophys-discuss.net/npg-2019-31/npg-2019-31-AC1-supplement.pdf

**Supplement:**

**Response to the First Referee.**

First, we would like to thank the Referee for her/his comments, all of which we have attempted to address. We think that the paper has been improved by them. Now we detail our response to each comment.

1. **There were no specific explanations for the choice of the set of parameters: nu, eta and N.**

   The choices are related to the findings in Domínguez et al. (2017), where we found that these values yield a good match between statistical features of the shell model and the fractal features of the *Dst* geomagnetic index. We have added explanations after Eq. (2) in this regard.

   The new text reads:

   *Based on Domínguez et al. (2017), where a comprehensive analysis of the statistical properties of the shell model for various values of $\nu$ and $\eta$ is carried out, we set $\nu = \eta = 10^{-4}$, as it is in the range where the model is able to best reproduce the intermittent behavior observed in magnetized plasmas. Given the values of the dissipative coefficients $\nu$ and $\eta$, we take $N = 19$, also consistent with the choices in Refs. Domínguez et al. (2017, 2018), a value which guarantees a nonlinear range sufficiently large to describe the system dynamics.*

2. **Magnetic dissipation rate is calculated but not the velocity dissipation rate. Why?**

   A velocity dissipation rate would be related to heating, which is not considered in our model. We have also added explanations after Eq. (3).

   The new text reads:

   *Notice that a dissipation rate for the velocity field can also be defined. However, this is not relevant to our model, as it would be related to heating, whereas there is no equation for temperature in our analysis. Magnetic storms, on the other hand, are related to magnetic dissipation rates.*

---

## Author Comment (AC2) · 26 Dec 2019

Please find enclosed the comments to the Referee concerning the paper titled "Study of an MHD Shell Model forced with Solar Wind data". The Referee's comments are in bold, and our answers in regular font. When- ever applicable, the modified portions of the text are included in italics in our response. We have also addressed the editorial comments regarding the figures. Finally, we would like to thank the Referee's comments. We think the paper is much improved thanks to them.

Regards,

Macarena Dominguez

Please also note the supplement to this comment:
https://www.nonlin-processes-geophys-discuss.net/npg-2019-31/npg-2019-31-AC2-
supplement.pdf
* * *
2019-31, 2019.

[Figure]

**Supplement:**

**Response to the Second Referee.**

First, we would like to thank the Referee for her/his comments, all of which we have attempted to address. We think that the paper has been improved by them. Now we detail our response to each comment.

1. **Introduction**
   **Other interesting applications of fractal dimension analysis on systems related to geophysical and biological extreme events can be found in Eftaxias et al. (2006, 2008). l. 48: please define the abbreviation "MHD" here.**

   We have added these references in the second paragraph of the Introduction.

   The new text reads:

   *. . . and the faults (Nanjo and Nagahama, 2004; Sahimi et al., 1993); in the study of various catastrophic events such as seismic and epileptic shocks, where the fractality of the relevant time series has been analyzed to extract information on precursor activity (Eftaxias et al., 2006, 2008); in music, . . .*

   We have also defined MHD where it first appears:

   The new text reads:

   *. . . computed in a magnetohydrodynamic (MHD) shell model integration. . .*

2. **Section 2**
   **l. 75: please define the abbreviation "GOY" here.**

   We have added the definition:

   The new text reads:

   *In this work, we use the MHD Gledzer-Ohkitani-Yamada (GOY) shell model*

3. **Results I**
   **In the right panel of Figures 3, 7 and 10 the fractal dimension curves are provided for the 13 years of solar cycle 23. However, the discrimination between curves corresponding to years of solar minimum and curves corresponding to years of solar maximum is not straightforward. In Figure 3 a couple of curves of solar minimum are mixed with the curves of solar maximum. Why? What are the years of solar minimum represented by these curves? Figure 7 discriminates between curves of solar minimum and maximum in a clearer way and Figure 10 does not discriminate at all.**

**Why is that?**

We acknowledge that Figs. 3, 7 and 10 were not very clear, and we have modified the manuscript to explain them in a better way. Their left panels had the same curves than their right panels, but colored in a different way, and we think this may have added to the confusion. Thus, we have only kept the left panel for the first figure, Fig. 3, in order to make the point, and then we avoid repeating the information in Figs. 7 and 10, which now have one panel only.
The text after Fig. 3 has been changed.

The new text reads:

*The left panel of Fig. 3 shows that there is a general trend for the fractal dimension to decrease with $j$. Previous results based on the shell model simulation (Domínguez et al., 2017) show that active and quiet states can be distinguished by their different behavior with $j$ (decreasing or increasing its fractal dimension for intermediate values of $j$, respectively). However, all curves in the left panel of Fig. 3 have the same trend, so it would seem that the fractal properties of the time series does not depend on the stage of the solar cycle.*

Regarding the mixing of solar minimum/maximum curves, this is simply showing that the fractal dimension is not a simple function of time. For instance, in Fig. 4 of Domínguez et al. (2018), the fractal dimension of the time series of magnetic field and velocity for the solar wind is calculated. Even though the fractal dimension for the magnetic time series has a clear trend of decreasing near solar maximum, the behavior is not monotonic. Changing $j$ adds more complexity to this, as the fractal dimension of the curve depends on the timescale of the sampling.
All these effects are found in Fig. 3. But it is interesting to notice, and we have attempted to emphasize it, that despite this, a general trend can be revealed, so that the fractal dimension during solar maximum is lower than during solar minimum.
Regarding Fig. 7, it does discriminate better. Fig. 3 is showing the fractality of the noise term $\mu(t)$ in Eq. (4), whereas Fig. 7 shows the fractality of the solution of Eq. (4), $\tilde{f}_n$. Thus, the Langevin equation is amplifying the differences in fractality between noise time series, but further studies on the Langevin equation are needed to understand this.
As to Fig. 10, it shows the fractality of the final output of the shell model. In Domínguez et al. (2018) it is explained how solar wind data are used to generate the forcing in the model. The same strategy is used in the present work, except that data are used to generate the noise term. As a result of that, the shell model is the result of a kind of average input for each year, and this may contribute to smooth the differences in fractalities in the output.
In any case, it is interesting that similar differences between fractalities of the solar wind and the magnetosphere were found in Domínguez et al. (2018) and Domínguez

et al. (2017), respectively.

4. **Results II**

**Figures 5 and 6 compare the variations of fractal dimension between solar wind data and dissipated magnetic energy: the fractal dimension of magnetic energy is higher than the fractal dimension of VBsouth during solar maximum. What's the interpretation of this observation? IMHO, it might mean that the complexity of the magnetosphere is higher than the complexity of the corresponding solar wind variations during solar maxima. Why is that? Back in 2006 there has been a study comparing the variations of the Hurst exponent (i.e., another complexity measure) between a geomagnetic activity index, indicating magnetic storm intensity, and VBsouth variations (Balasis et al., 2006). This was done for 2001 a year around solar cycle maximum. It was found that while the magnetosphere is mostly driven by the solar wind the critical feature of persistency (lower complexity) in the magnetosphere is the result of a combination of solar wind and internal magnetospheric activity rather than solar wind variations alone. In that study the degree of complexity between Dst and VBsouth was apparently different. In the present study the same seems to hold for the dissipated magnetic energy and VBsouth. Please comment upon the past and present findings.**

First, we point out that Figs. 5 and 6 do not compare solar wind and magnetic energy fractalities, as the fractal dimension in those figures, the black line, is always the same and corresponds to $\mu(t)$ (solar wind).

In order to compare solar wind and magnetic energy, we need to compare Fig. 10, with either Fig. 3 or 7, and the analysis shows that complexity in the magnetosphere is lower than in the solar wind. We do thank the Referee for pointing out the reference of Balasis et al. (2006) to us. We think that the comparison between both works is not straightforward, given that different timescales are considered, and different complexity measures are employed. On the other hand, this highlights the possible complications in the analysis of complexity measurements, specially because, given the complex dynamics of the system considered, different metrics may yield different information, and further work should be done in order to better understand how such metrics add to the general picture.

We have added two paragraphs, at the end of Sec. 6, to comment on this.

The new text reads:

*It is interesting to discuss Figs. 10, 3, and 7, in the light of comparative studies of complexity in the solar wind and the magnetosphere, although this work attempts to make a very simplified model of the interaction between the solar wind and the Earth's magnetosphere. In effect, Figs. 3 and 7 show the complexity of the drivers of the shell*

*model, which we may loosely associate with the solar wind driving the magnetosphere, whereas Fig. 10 shows the complexity of the output of the shell model, which may represent the magnetospheric activity, following the analogy.*

*Thus, these plots show that the complexity of the driven system (Fig. 10) is more similar to the complexity of the driver (Figs. 3, 7) during solar maximum, than during solar minimum; and that the complexity of the driven system is typically lower than the complexity of the driver. This is different from results in Balasis et al. (2006), where a study in terms of Hurst exponents was made, finding that complexity in the magnetosphere is larger than in the solar wind. However, it should also be noticed that in our work, longer term trends are studied (1-year windows), instead of timescales of the order of the duration of geomagnetic storms. The different timescales, and the use of different metrics for complexity, could be relevant to compare both results.*

5. **Conclusions**

   **Please provide motivation / rationale for performing this study. What are the differences / improvements between the present study and recent studies on the topic by the same group of authors (e.g. Dominguez et al., 2018)? What is the significance of the obtained results for the Space Weather community? Eventually, please compare your results with other pertinent studies (e.g. Donner et al., 2019).**

We thank the Referee for the suggestion, and we have added three paragraphs at the end of Sec. 7 to comment on this.

The new text reads:

*Despite this, it is interesting to see that some results are consistent with previous studies, based directly on data. Fractal dimensions in Figs. 3 and 7 measure the complexity of the drivers of the shell model, which we may loosely associate with the solar wind driving the magnetosphere, whereas Fig. 3 measures the complexity of the output of the shell model, which may represent the magnetospheric activity, following the analogy. Results are similar to those calculated for the solar wind (Domínguez et al., 2018) and for the Dst index (Domínguez et al., 2014), in the sense that values of the fractal dimensions suggest that the complexity of the solar wind is larger that the complexity of the magnetosphere, mesured using the same box-counting approach as presented here.*

*Given the complex dynamics in the system studied, we should not expect that a single metric contains all the information, and thus results may depend on the method used. For instance, Hurst exponents are used in Balasis et al. (2006), which suggests that complexity in the magnetosphere is larger than in the solar wind. On the other hand, timescales observed are also different in our work and in Balasis et al. (2006), and this can also be relevant to evaluate complexity in a physical system.*

*Nevertheless, it is interesting that various studies have considered the use of fractal dimensions, using several strategies, as a means to extract information on solar wind-magnetosphere interaction, either in the sense of precursor activity (Donner et al., 2018; Balasis et al., 2006), or longer-term trends (Domínguez et al., 2014, 2018). simulation-based studies may help to understand to what extent complexity measures may be relevant for this task.*